# AI-Assisted Image Recognition of Cervical Spine Vertebrae in Dynamic X-Ray Recordings

**DOI:** 10.3390/bioengineering12070679

**Published:** 2025-06-20

**Authors:** Esther van Santbrink, Valérie Schuermans, Esmée Cerfonteijn, Marcel Breeuwer, Anouk Smeets, Henk van Santbrink, Toon Boselie

**Affiliations:** 1Department of Neurosurgery, Maastricht University Medical Centre, 6229 HX Maastricht, The Netherlands; valerie.schuermans@mumc.nl (V.S.); anouk.smeets@mumc.nl (A.S.); h.van.santbrink@mumc.nl (H.v.S.); t.boselie@mumc.nl (T.B.); 2Department of Neurosurgery, Zuyderland Medical Centre, Henri Dunantstraat 5, 6419 PC Heerlen, The Netherlands; 3CAPHRI Institute for Public Health and Primary Care, Maastricht University, 6200 MD Maastricht, The Netherlands; 4Department of Biomedical Engineering, Eindhoven University of Technology, 5600 MB Eindhoven, The Netherlands; e.e.j.cerfonteijn@student.tue.nl (E.C.); m.breeuwer@tue.nl (M.B.); 5Department of Electrical Engineering, Eindhoven University of Technology, 5600 MB Eindhoven, The Netherlands

**Keywords:** cervical spine, qualitative motion analysis, image segmentation, U-Net, X-ray recording, deep-learning model, motion pattern

## Abstract

*Background:* Qualitative motion analysis revealed that the cervical spine moves according to a consistent pattern. This analysis calculates the relative rotation between vertebral segments to determine the sequence in which they contribute to extension, demonstrating a mean sensitivity of 90% and specificity of 85%. However, the extensive time that is required limits its applicability. This study investigated the feasibility of implementing a deep-learning model to analyze qualitative cervical motion. *Methods:* A U-Net architecture was implemented as 2D and 2D+t models. Dice similarity coefficient (DSC) and Intersection over Union (IoU) were used to assess the performance of the models. Intraclass Correlation Coefficient (ICC) was used to compare the relative rotation of individual vertebrae and segments to the ground truth. *Results:* IoU ranged from 0.37 to 0.74 and DSC ranged from 0.53 to 0.80. The ICC scores for relative rotation ranged from 0.62 to 0.96 for individual vertebrae and from 0.28 to 0.72 for vertebral segments. For segments, 2D+t models presented higher ICC scores compared to 2D models. *Conclusions:* This study showed the feasibility of implementing deep-learning models to analyze qualitative cervical motion in dynamic X-ray recordings. Future research should focus on improving model segmentation by enhancing recording contrast and applying post-processing methods. Improved segmentation accuracy will enable routine use of the analysis of motion patterns in clinical research. The absence or presence of a motion pattern, or identification of new patterns has the potential to aid in clinical decision-making.

## 1. Introduction

A clear definition of physiological cervical spine motion is lacking. Segmental range of motion (sROM) has been proposed as a metric to evaluate cervical spine motion. However, sROM is limited by high intra- and interindividual variability and restricted to static endpoints [1,2]. The sequence of segmental contribution (SSC) is an alternative metric that is used to analyze qualitative motion of the cervical spine [3]. To determine SSCs, relative rotations of the segments are tracked in dynamic X-ray recordings. The SSC is a consistent metric and has a mean sensitivity of 90% and specificity of 85% to differentiate motion patterns between asymptomatic individuals and patients with a radicular syndrome [4]. In a cross-validation study, two individual methods to analyze cervical spine motion in dynamic X-ray recordings demonstrated excellent agreement with a median difference below 0.2 degrees [5]. Motion pattern analysis revealed that various segments contribute at different moments during the extension movement of the cervical spine.

In young asymptomatic individuals, a consistent motion pattern of vertebrae C4 to C7 is observed during extension [4]. The pattern is defined as the order of maximum contribution of each motion segment to the extension movement, i.e., a peak in the relative rotation with respect to the cumulative amount of rotation of vertebrae C4 to C7. An RCT comparing anterior cervical discectomy with arthroplasty (ACDA) to anterior cervical discectomy (ACD) in patients with radicular syndrome demonstrated the impact of surgery on observed motion patterns. Before surgery, the consistent motion pattern that was previously defined in asymptomatic individuals was present in 37.5% of patients. In the ACDA group, in 80% of patients this same motion pattern was restored one year after surgery, compared to 20% in the ACD group [6]. The difference in motion patterns between the groups can be explained by the presence of fusion in the ACD group, and maintenance of movement in the ACDA group. However, in a subsequent study, we observed that the motion pattern was absent in both treatment groups after a mean follow-up of 11 years. The range of motion (ROM) was however preserved in the ACDA group compared to the ACDF group in this long-term follow-up [7]. It has already been shown that ROM decreases with age due to the natural aging process. Qualitative analysis of cervical motion in elderly showed that the consistent motion pattern that was found in young asymptomatic individuals was absent [8]. The presence or absence of a consistent motion pattern among different groups raises questions and requires further investigation of cervical spine motion. In clinical practice, the presence of a consistent motion pattern might play a role in clinical decision-making such as choosing surgical strategies.

Qualitative motion analysis has not been widely applied in research or clinical practice due to its time-consuming nature and the need for trained and experienced individuals. To increase feasibility, artificial intelligence (AI) algorithms are implemented to design deep-learning models for the segmentation of cervical vertebrae. Promising results have been reported in studies that use these models to recognize cervical injuries on lateral X-rays, with accuracy for segmentation of cervical vertebrae ranging from 92 to 99% [9,10]. One study focused on calculating ROM of cervical vertebrae in static lateral flexion and extension radiographs and showed good accuracy [11].

In contrast to static X-rays, dynamic X-ray recordings can incorporate a temporal dimension, enabling a 2D + time (2D+t) model. Implementation of both a U-Net and capsule network (CapsNet) for 2D, 2D+t and 3D model approaches to segment brain structures in MRI data showed comparable performance in the segmentation of the third ventricle [12]. The 3D U-Net achieved the highest segmentation accuracy of 96% Dice score, a metric that measures the overlap between the predicted segmentation and the ground truth. The segmentation accuracy of the 2D+t U-Net was similar to the results of the 2D U-Net with a Dice score of 91% and 90%, respectively. Similar findings were observed when comparing a U-Net with a SegNet architecture for the 2D, 2D+t and 3D model approaches [13]. Five medical datasets were investigated, both CT and MRI data, and the segmentation performance of the 2D+t approach did not improve compared to both 2D and 3D. The effectiveness of the U-Net architecture for segmentation using a 2D+t approach has been demonstrated, but conclusions on the efficacy of 2D+t versus 2D methods varied.

To enable qualitative analysis of the cervical spine, this study aimed to develop a 2D and 2D+t deep-learning model, trained on previously annotated dynamic X-ray recordings, to automatically calculate the relative rotation of each vertebra.

## 2. Materials and Methods

### 2.1. Population

Previous studies by our group investigated motion patterns in healthy, asymptomatic individuals in different age groups. Detailed information can be found in the published full-text articles [4,6,7,14,15]. Manually annotated data from these studies were combined for the training of this algorithm. Dynamic X-ray recordings were all made following the same protocol. Participants were seated on a crutch, adjustable in height, with their neck, shoulders, and head free. The subjects were asked to move their heads from maximal extension to maximal flexion, and vice versa, as fluent as possible and without moving the upper part of his body in about 10 s, using a metronome. While making the recordings the participants shoulders are kept as low as possible to ensure that all the cervical vertebrae are visible. The recordings were either made with the Siemens Arcadis Avantic VC10A (Siemens AG, Munich, Germany), Toshiba Infinix Vi (Ōtawara-shi, Tochigi-ken, Japan), or Philips Allura Xper FD20 X-ray system (Best, The Netherlands), capturing frames of 1024 × 1024 pixels at 15 frames per second. Images were not compressed.

### 2.2. Manual Annotation

Template areas containing each vertebral body and the skull base were manually drawn by two trained individuals (TB and VS) in the median frame of the recording, labeled C0–C7. Two-way mixed ICCs were calculated to assess reproducibility and consistency. The bottom part of the sella turcica and clivus are projected in the X-ray image as a stable and structure rich midline area of the skull. This is therefore an ideal template to track the movement of C0 (the skull base). Image recognition software, based on matching normalized gradient field images on a best-fit principle, was then used to locate the vertebrae in the other frames of the recordings, as described by Reinartz et al. [16]. The contours of the vertebra were manually checked and if necessary adjusted to fit the corresponding vertebrae in every frame of the recording (Figure A1, Appendix A). These image recognition-assisted manual annotations were considered the ground truth.

### 2.3. Development of the Model

Two image dimensions were evaluated to determine the best approach for analyzing motion patterns in clinical research. To choose a uniform image size for both image dimensions, the maximum width and height of the manual annotations of vertebrae C1–C7 were determined in each frame across all recordings. The maximum Region of Interest (RoI) that centered around the vertebrae in a single frame was identified to be 608 × 505 pixels. To capture the entire spine movement in a single view, the absolute minimum and maximum x- and y-coordinates of the vertebrae in each recording were used to determine the RoI of 826 × 543 pixels. To ensure a margin around the vertebrae and optimal visibility of vertebra C0, the two image dimensions were set to 640 × 640 pixels and 832 × 576 pixels, respectively.

2D models were compared to 2D+t models. The 2D+t models integrated temporal information with the hypothesis that the predicted vertebral shape will become more consistent over time, leading to more accurate relative rotation values. The neural network that was used for the segmentation task of this study was a five layered U-Net, widely used in medical segmentation studies [17,18]. For the 2D model, each frame was presented as input to the model. For the 2D+t model, multiple frames were presented as input to the model. Consecutive frames were given as input to the model, with the segmentation only generated on the middle frame. All models have nine output channels, one for each of the eight vertebrae and one for the background class. To determine the ideal number of frames for the 2D+t models, an ablation study was performed. As the application was not needed in real-time, it was possible to use the frames acquired before and after frame *t*, e.g., *t* − 1 and *t* + 1 in case of three input frames. As pre-processing step, both image sizes were normalized with the mean and standard deviation of pixel values of the training subset. On the training dataset, data augmentation was applied to artificially increase the diversity of the training data by applying various transformations. Specifically, Gaussian noise was introduced, and the contrast was adjusted. The hyperparameters settings for the model training were the Adam optimizer with a learning rate of 0.0001, the weighted Dice loss function (Equation (1)), a batch size of eight images, and for a maximum of 300 epochs. The weighted loss function is a strategy to increase the significance of the vertebrae, by determining the ratio of foreground pixels (*g*) to the total amount of pixels (*N*) to use as weights (Equation (2)).(1)$$L_{Dice}=1-\frac{2\cdot \sum_{i}^{N}p_{i}\cdot {gt}_{i}}{\sum_{i}^{N}p_{i}^{2}+\sum_{i}^{N}{gt}_{i}^{2}}$$
where *p_i_* represents the predicted probability of pixel *i* and *gt_i_* represents the ground truth of pixel *i*.(2)$$W=\frac{1}{{\left(\sum_{i=1}^{N}g_{i}\right)}^{2}},$$

For both 2D and 2D+t models, the two dimensions were analyzed, resulting in four different models, models A to D. An overview of models A to D can be found in Table 1.

The output of the model, given in probability values, was transformed into a binary mask by applying an optimized threshold between 0.1 and 0.9 (Figure A2, Appendix A). The optimal threshold for each vertebra was determined via the Precision-Recall (PR) curve, as it addresses the class imbalance by focusing on foreground pixel performance. For thresholds ranging from 0.1 to 0.9 in steps of 0.1, the PR-curve was generated, and the optimal threshold was selected based on the highest F1-score, calculated from the harmonic mean of precision and recall derived from the PR-curve. The threshold varied between vertebrae per model (Table A1, Appendix A). With the optimal threshold-vertebra combinations, the Intersection over Union (IoU) and Dice score were computed for each vertebra in every frame for each trained model.

### 2.4. Dataset

Dynamic X-ray recordings with annotated and corrected contours were available for 39 healthy controls. Extension recordings were made at two time points, resulting in 78 extension recordings. In eleven cases, flexion recordings were also analyzed, resulting in a total of 89 recordings in these 39 subjects. Each recording consisted of ±52 individual frames that were analyzed. Altogether this resulted in a total of 4671 frames. The recordings were split into subsets for training, validation, and testing, with a split of approximately 55–20–25%, respectively (Table 2). These percentages are commonly used as a standard distribution for deep-learning models, resulting in 2709 frames for training, 964 frames for validation, and 998 frames for testing. Recordings were allocated per individual into one of the subsets.

### 2.5. Generation of Mean Shape

The model predictions were made per individual frame of a recording, resulting in a slightly different shape in each frame, which could be caused by overprojection of the 3D structures in the 2D plane or due to coupled motion. However, vertebrae are rigid bodies, and thus their actual shape does not vary throughout the movement. To include this physical property, a mean shape was created for each vertebra per recording to calculate the relative rotation.

To quantify the overlap between the mean shape and the ground truth, the Intersection over Union (IoU) was calculated. Similarly, the IoU was calculated between the mean shape and the model predictions.

### 2.6. Outcome Measures

To assess the performance of each model, the IoU and the Dice similarity coefficient (DSC) were calculated.

The IoU measures the overlap between the predicted segmentation and the ground truth by calculating the ratio of the intersection area to the union area (Equation (3)). Typically, IoU values above 0.7 are deemed good, those between 0.5 and 0.7 indicate moderate overlap, and values below 0.5 suggest poor overlap:*IoU (A,B) = (|A ∩ B|)/(|A ∪ B|)*,(3)

The DSC is also a measure of overlap between the predicted segmentation and the ground truth, by dividing twice the size of the intersection area with the sum of their individual sizes (Equation (4)) [19]. A DSC is considered good when above 0.7, a DSC between 0.5 and 0.7 is considered moderate, and below 0.5 is considered poor [20]. The main difference between IoU and DSC is the fact that under- and over-segmentation is penalized more by the IoU.*DSC (A,B) = (2 × |A ∩ B|)/(|A| + |B|)*,(4)

The primary outcome will be the intraclass correlation coefficient (ICC) of the relative rotation of individual vertebrae over time. The relative rotation values will be calculated using a mean shape of each vertebra per recording. The mean shape was positioned on the centroid of the model prediction on each frame and subsequently rotated iteratively until maximal overlap was achieved. The fitted mean shape was used to calculate the rotation of each vertebra in the consecutive frames of the entire recording (*dθ_C_*), as shown in Equation (5), where *θ* denotes an angle in frame *t*.*dθ_C_ = θ_C(t)_ − θ_C(t−1)_*,(5)

On the first frame of the recording *dθ_C_* = 0 as there was no preceding frame (*t* − 1). For the 2D+t models, *dθ_C_* = 0 for frames that were not considered as a middle frame, such as the first and last frames of the recording in case of three input frames. *C* indicates one of the eight vertebrae.

The secondary outcome measure was the ICC of the relative rotation of vertebral segments over time. Relative rotation for segments was also calculated using a mean shape. With the relative rotation values of each vertebra, the relative rotation between two vertebrae (*dθR*) was calculated via Equation (6).*dθ_R_ = dθ_Ck_ − dθ_Cl_*,(6)
with *Ck* and *Cl* each indicating another vertebra.

The Wilcoxon Signed Rank test was used to assess significant differences in outcome performance between the different models. A *p*-value ≤ 0.05 was considered statistically significant.

## 3. Results

### 3.1. Ablation Study 2D+T

The ablation study of the 2D+t models was conducted using three, five, seven, and nine input frames. It was observed that the evaluation scores did not further improve when comparing nine input frames to seven. For both model configurations, seven input frames most frequently resulted in the significantly highest score. Results of the ablation study for Model C and Model D can be found in Appendix A, Table A2 and Table A3.

### 3.2. Model Performance

The mean IoU score ranged from 0.37 to 0.72 for model A, from 0.51 to 0.72 for model B, from 0.45 to 0.74 for model C, and from 0.49 to 0.72 for model D. The mean DSC ranged from 0.53 to 0.82 for model A, from 0.66 to 0.82 for model B, from 0.61 to 0.83 for model C, and from 0.64 to 0.82 for model D. The mean IoU and DSC for each vertebra across all models are presented in Table 3, with the highest score indicated in bold. An asterisk denotes a significant difference between the highest score and the scores from the other three models.

Figure 1 illustrates the segmentation of each vertebra created by model C comparing the first, median, and final frames of the recording to the ground truth. From top to bottom, good, average, and poor model segmentation results are displayed compared to the ground truth. Equal results were observed for models A, B, and D, as seen in Figure A3, Figure A4 and Figure A5, Appendix A.

### 3.3. Mean Shape

The mean shape was generated for vertebrae C1–C7 for each model option, excluding the skull base C0 because of its variable shape. For several recordings per model, it was not possible to create a mean shape due to deviations in the model segmentation. Figure 2 displays the mean shape of C5 compared to the ground truth and the model segmentation. The mean shape that was generated for the other vertebrae can be seen in Figure A6 and Figure A7, Appendix A.

To compare the accuracy of the mean shape generated by the model to the ground truth, the IoU was calculated for each vertebra and model, presented in Table 4. The IoU score ranged from 0.56 to 0.80 for model A, from 0.56 to 0.84 for model B, from 0.61 to 0.84 for model C, and from 0.61 to 0.84 for model D.

### 3.4. ICC of Relative Rotation of Individual Vertebrae

The ICC score for individual vertebrae ranged from 0.819 to 0.962 for model A, from 0.798 to 0.948 for model B, from 0.683 to 0.907 for model C, and from 0.620 to 0.878 for model D, presented in Table 5. The highest mean score between the models is indicated in a red bold font. The number of recordings on which the ICC is based is also presented in Table 5. Figure 3a illustrates the trajectory of C1 Model A with a high ICC (0.990), and Figure 3b illustrates the trajectory of C3 Model D with a low ICC (0.298).

### 3.5. ICC of Relative Rotation of Vertebral Segments

The ICC score for the vertebral segments ranged from 0.511 to 0.685 for model A, from 0.408 to 0.627 for model B, from 0.382 to 0.770 for model C, and from 0.281 to 0.724 for model D, presented in Table 6. The highest mean score between the models is indicated in a red bold font. The number of recordings on which the ICC is based is also presented in Table 6. Figure 4a illustrates the trajectory of C1–C2 Model D with a high ICC (0.890), and Figure 4b illustrates the trajectory of C5–C6 Model B with a low ICC (0.000).

## 4. Discussion

The results of this study show the feasibility of implementing a 2D and 2D+t deep-learning model to recognize cervical vertebrae in dynamic X-ray recordings. Based on acceptable ICC scores for the relative rotation of individual vertebrae, the investigated models can accurately track vertebrae over time. To calculate the segmental relative rotation, the models need to be improved further, as the current results are still poor to moderate.

In the ablation study for the 2D+t models, an improvement in the IoU score was observed when seven frames were compared to three or five frames. In the current literature, three or five frames are often used in static imaging [12,21,22]. As cervical spine motion can be observed between frames, a model that analyzes the temporal dimension by comparing consecutive frames can benefit from additional information. However, using nine frames did not lead to further improvement in the IoU score, indicating that too much information away from the median frame leads to a poorer outcome.

The IoU results of the different models are better overall for C1 to C4 compared to C0 and C5 to C7. Figure 1, Figure A3, Figure A4 and Figure A5 also display that the lower vertebrae are more fragmented. This can be explained by the different features of the cervical vertebrae and the imaging technique. Vertebrae C1 and C2 each have a unique shape that facilitates easier prediction by the model. In contrast, vertebrae C3–C7 exhibit more similar features. In particular, vertebrae C5 and C6 pose a challenge for the model, as they are frequently misclassified due to their strong resemblance [23]. The segmentation accuracy of the skull base C0 is compromised due to the absence of a visual feature at the position of the straight line in the ground truth, especially when C0 is partially out of view in certain frames of the recording. Due to the imaging technique, C6 and C7 are often obstructed by the shoulders, which makes it challenging for the model to accurately predict these vertebrae. These challenges may reduce model performance in cases where precise segmentation is essential for angle determination.

Inaccuracies in model segmentation can to some extent be explained by the quality of recordings. In multiple recordings, the contrast between the vertebrae and background is less pronounced, which leads to worse outcomes. In contrast, recordings with distinct edges of the vertebrae and enhanced contrast lead to improved outcomes. In some included recordings, increased tube voltage was necessary to enhance visualization of vertebrae C6 and C7 located behind the shoulders, albeit at the expense of overexposure to the other structures, resulting in light grey vertebrae with minimal contrast against the background. These findings provide valuable knowledge for the acquisition of future X-ray recordings. Examples of recordings with low and good results are presented in Figure A8 and Figure A9, Appendix A.

Even though the shape of vertebral bodies is rigid, model segmentation shows different shapes of the vertebral bodies over time. This variance in segmentation is caused by overprojection of structures on the recording. The generated mean shape for all vertebrae had an acceptable IoU compared to the ground truth, presented in Table 4. Similarly to the model segmentation, C5 to C7 demonstrated lower IoU scores. This emphasizes the complexity for the model to accurately segment the lower vertebrae. The proposed solution for the different shapes, creating a mean shape, led to better results for the ICC scores, as presented in Table 6 and Table A4.

The ICC scores for the relative rotation of individual vertebrae are similar between the 2D and 2D+t models. The mean ICC scores ranged from good to excellent, indicating that all models can accurately track individual vertebrae over time. The ICC scores for relative rotation of vertebral segments are lower. This can be explained by comparing the overall rotation of an individual vertebra to a segment. Boselie et al. calculated that between frames a segment can rotate as little as 0.3 degrees, less than the measurement error [4]. Rotation of individual vertebrae between frames ranges from 2 to 3.5 degrees, illustrated in Figure 3a,b. Therefore, rotation over time of individual vertebrae is less sensitive to measurement errors or segmentation inaccuracies. Additionally, inaccuracies in segmentation of vertebrae above or below do not affect the measured relative rotation of the individual vertebra.

When the 2D and 2D+t models are compared to track vertebral segments over time, the 2D+t results in higher ICC values, confirming that incorporating a time dimension will lead to a more consistent segmentation shape than the 2D models. We also analyzed two image sizes to measure if a more centralized frame led to more accurate model segmentation. Between the image sizes, we did not observe a relevant difference in our outcomes, indicating that both can be used for qualitative cervical motion analysis.

A limitation of this study is the angular shape of the ground truth segmentation, as the natural shape of a vertebra is smoother. The ground truth is created with the algorithm of Reinartz et al., which is already very time-consuming [16]. Adding a greater number of points to achieve a smoother and more realistic shape would further extend the processing time. The difference in shape can lead to lower IoU or DSC scores. It is favorable that the model segmentation already creates a more natural looking vertebra.

Another limitation is that the 2D recordings capture only the sagittal rotation. Other motion planes, e.g., axial rotation or lateral flexion, have not been analyzed. To address these limitations, future work could explore the integration of multi-modal imaging data to improve vertebral segmentation, especially in complex cases. Incorporating axial rotation information by combining biplanar X-ray with CT or MRI could enhance the model’s ability to capture three-dimensional motion patterns, which are clinically relevant for assessing cervical spine kinematics.

Lastly, the number of recordings included in ICC calculation is limited, and we only reported the minimum and maximum values. Some ICC values are based on one or two recordings. Therefore, these values should be interpreted with caution. For each level, the recording was excluded if the mean ICC was negative or if the recording contained at least one outlier. A negative value occurs when the ground truth segmentation moves opposite to the model segmentation. An outlier occurs when the threshold, based on the minimum and maximum raw values of the gold standard, is exceeded. The most probable cause of an outlier is the model segmentation used to fit either the mean shape or the model segmentation in the subsequent frame. Deviating shapes and differences in shape between consecutive frames can result in sub-optimal fits during rotation. Also, most outliers were detected for C5 and C6. The shape of these vertebrae is very similar, making it more complex for the model to differentiate between C5 and C6. We decided to exclude these recordings because the outliers originate from model segmentation issues.

For future use, model segmentation can be improved by several methods. One method is improving the quality of the recordings by enhancing the contrast. This can be achieved by histogram equalization or histogram matching [24]. Another method is training the model to recognize the vertebra in a specific sequence. Incorporating a sequence will prevent the model from interchanging vertebrae. Also, post-processing techniques can be applied when the model segmentation is fragmented. Currently, only the largest segment is selected for further analysis in the mean shape and relative rotation calculations. By incorporating post-processing methods, fragmented segments can be merged into a single segmentation using techniques like morphological operations, e.g., dilation and closing, and majority voting for consistent foreground pixels across multiple frames [25]. Another method is to recognize frames that cause outliers and replace these frames with the preceding or following frame. This is especially relevant for recordings with a limited number of outliers. Improving the model segmentation in each frame with post-processing techniques will result in a more accurate mean shape and more precise relative rotation values, ultimately improving the ICC scores. With improved segmentation, the model can be used for measuring other metrics, such as the cervical lordosis and T1 slope. It has already been shown to be feasible to measure this in static X-rays [26]. Analyzing these metrics in dynamic recordings will provide further insights into the normal movement of the cervical spine.

This model can accurately calculate the relative rotation of an individual vertebra. However, the relative rotation analysis of vertebral segments needs to be further improved to enable motion pattern analysis. Such motion patterns have previously been shown to allow differentiation between asymptomatic individuals and patients with cervical radiculopathy. More clinically relevant applications are likely, particularly in evaluating spinal instability, degenerative changes over time, and post-surgical biomechanics. For instance, in case segmental motion is preserved preoperatively, a motion-preserving device may be considered over fusion in anterior cervical discectomy procedures. Conversely, the absence of motion may justify a fusion approach. Additionally, routinely analyzing changes in motion patterns before and after surgery could yield insight into the development of complications such as adjacent segment disease.

By leveraging big data in combination with these motion analyzes, previously unrecognized motion patterns that might influence clinical outcome and possible future treatment strategies could be identified. These applications underscore the potential of AI-based motion tracking not only as a diagnostic tool but also as a future individualized decision-support system in cervical spine surgery.

## 5. Conclusions

Implementing a 2D or 2D+t deep-learning model to analyze the relative rotation of individual vertebrae leads to accurate results compared to the ground truth. For the calculation of the relative rotations of vertebral segments, the 2D+t models, incorporating a temporal dimension, seem to do better than 2D models. However, these results remain poor and require further improvement. Future research should focus on improving the model segmentation by enhancing contrast and applying post-processing methods. This will enable motion pattern analysis in large groups. Additionally, integrating multi-dimensional or multi-modal data and applying the model to clinical datasets could help to increase the robustness and clinical relevance of the approach. Ultimately, these advancements could enable the analysis of motion patterns in larger and more diverse patient groups, forecasting long term clinical outcome and, for the far future, serving as a support for more personalized patient care.

## Figures and Tables

**Figure 1 bioengineering-12-00679-f001:**
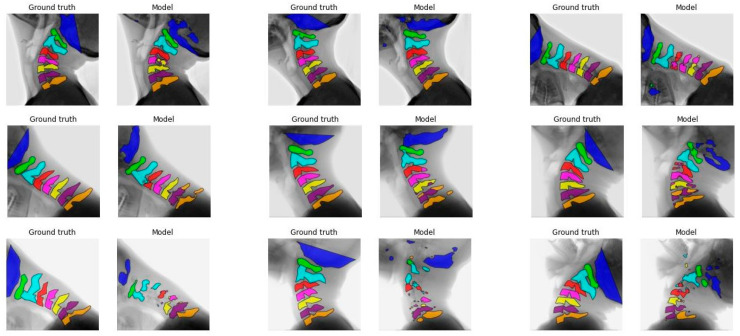
Comparison between the ground truth and the segmentation made by model C. Each vertebra is indicated with another color. Each row depicts a recording of a different test subject. In the first row, the recording with the best visual results is shown; in the second row, the median visual accuracy is displayed, and the last row shows the worst visual result. The first column shows the first frame of the recording, the middle column shows the median frame of the recording, and the last column shows the last frame of the recording. The first row depicts a flexion recording, while the second and third rows show extension.

**Figure 2 bioengineering-12-00679-f002:**
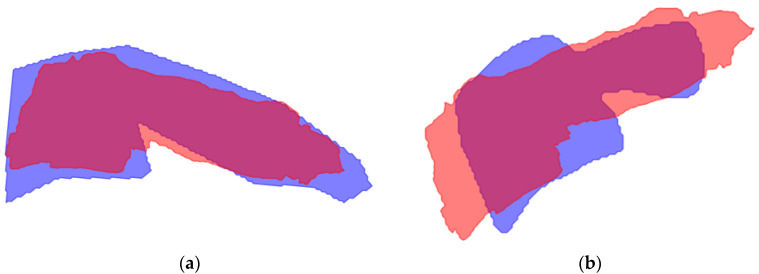
Visualization of the mean shape of C5. Both (**a**) and (**b**) are calculated on the median frame. (**a**) The ground truth is blue, the mean shape is red, and overlap is displayed in purple; (**b**) The model segmentation is blue, the mean shape is red, and the overlap is displayed in purple.

**Figure 3 bioengineering-12-00679-f003:**
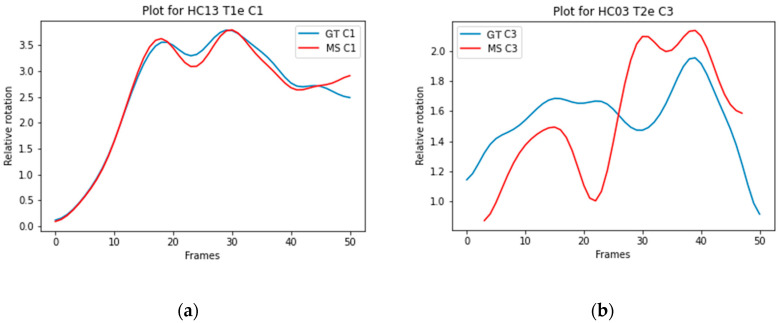
(**a**) Graphical representation of the C1 trajectory in model A (ICC = 0.990), comparing the ground truth (blue line) with the mean shape (red line). The *x*-axis shows the vertebral rotation in degrees, and the *y*-axis indicates the number of frames; (**b**) Graphical representation of the C3 trajectory in model D (ICC = 0.298), comparing the ground truth (blue line) with the mean shape (red line).

**Figure 4 bioengineering-12-00679-f004:**
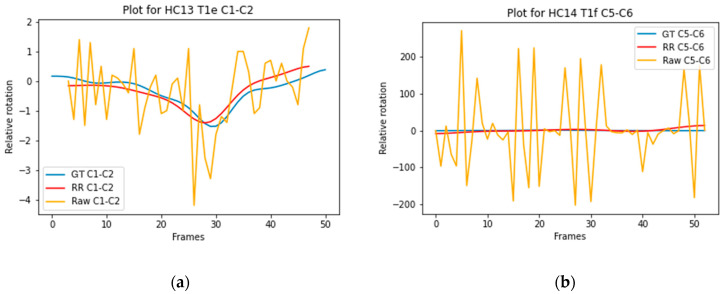
(**a**) Graphical representation of the C1–C2 trajectory in model D (ICC = 0.890), comparing the ground truth (blue line) with the mean shape (red line) and the unfiltered mean shape trajectory (yellow line). The *x*-axis shows the relative rotation of the vertebral segment in degrees, and the *y*-axis indicates the number of frames; (**b**) Graphical representation of the C5–C6 trajectory in model B (ICC = 0.000), comparing the ground truth (blue line) with the mean shape (red line) and the unfiltered mean shape trajectory (yellow line).

**Table 1 bioengineering-12-00679-t001:** Overview of the tested models A to D.

Model	Dimension	
A	640 × 640	2D
B	832 × 576	2D
C	640 × 640	2D + time
D	832 × 576	2D + time

**Table 2 bioengineering-12-00679-t002:** Overview of the included data divided into subsets, displayed for number of individuals and number of recordings.

Data Subset	Individuals (*n* =)	Recordings (*n* =)
Training (55%)	21	52
Validation (20%)	8	18
Testing (25%)	10	19

**Table 3 bioengineering-12-00679-t003:** Performance of model options A–D on the test set.

	IoU				DSC			
Model	A	B	C	D	A	B	C	D
C0	0.37	**0.51 ***	0.45	0.49	0.53	**0.66 ***	0.61	0.64
C1	0.71	**0.72 ***	**0.72**	0.7	0.81	**0.82 ***	0.82	0.81
C2	**0.72**	0.71	**0.72**	**0.72**	**0.82**	0.81	**0.82**	**0.82 ***
C3	0.7	0.72	**0.74 ***	0.72	0.8	0.82	**0.83 ***	0.82
C4	0.6	**0.64 ***	0.63	0.63	0.72	**0.76 ***	0.74	0.75
C5	0.51	0.56	0.52	**0.58 ***	0.64	0.69	0.64	**0.71 ***
C6	0.51	0.55	0.57	**0.59 ***	0.65	0.69	0.7	**0.73 ***
C7	0.51	0.52	**0.55 ***	0.54	0.65	0.66	**0.68 ***	0.67

IoU = Intersection over Union, DSC = Disc Similarity Coefficient. The highest IoU and DSC values for each vertebra are indicated in bold. Significant differences compared to all other options are marked with an asterisk.

**Table 4 bioengineering-12-00679-t004:** The mean IoU score calculated between the ground truth and mean shape of each vertebra, per model A–D.

	A	B	C	D
C1	0.76	0.76	0.78	0.75
C2	0.80	0.79	0.78	0.76
C3	0.79	0.84	0.84	0.84
C4	0.69	0.81	0.78	0.75
C5	0.56	0.62	0.61	0.61
C6	0.60	0.56	0.63	0.66
C7	0.63	0.63	0.62	0.56

IoU = Intersection over Union.

**Table 5 bioengineering-12-00679-t005:** ICC scores individual vertebrae calculated per model using the mean shape.

	Model A		Model B		Model C		Model D	
Vertebra	ICC [min–max]	*n*	ICC [min–max]	*n*	ICC [min–max]	*n*	ICC [min–max]	*n*
C1	**0.962** [0.916–0.993]	7	0.948 [0.834–0.996]	13	0.888 [0.471–0.997]	12	0.843 [0.479–0.982]	12
C2	**0.904** [0.699–0.996]	10	0.882 [0.449–0.978]	12	0.868 [0.413–0.988]	12	0.796 [0.400–0.985]	12
C3	0.871 [0.422–0.993]	7	**0.917** [0.826–0.976]	9	0.741 [0.132–0.979]	7	0.620 [0.298–0.909]	6
C4	0.880 [0.814–0.960]	3	0.812 [0.601–0.927]	7	**0.907** [0.899–0.923]	3	0.636 [0.343–0.820]	3
C5	**0.904** [n/a]	1	0.798 [0.650–0.945]	2	0.683 [0.658–0.680]	2	0.775 [0.707–0.864]	3
C6	**0.982** [n/a]	1	0.830 [0.665–0.995]	2	0.769 [0.471–0.979]	4	0.878 [0.639–0.966]	8
C7	0.819 [0.732–0.905]	2	0.869 [0.650–0.954]	5	**0.879** [0.785–0.974]	5	0.863 [0.697–0.942]	4

ICC = Intraclass Correlation Coefficient. Per vertebra the highest mean score is indicated in red bold.

**Table 6 bioengineering-12-00679-t006:** ICC scores for vertebral segments calculated per model using the mean shape.

	Model A		Model B		Model C		Model D	
Segment	ICC [min–max]	*n*	ICC [min–max]	*n*	ICC [min–max]	*n*	ICC [min–max]	*n*
C1–C2	0.685 [0.481–0.988]	5	0.627 [0.136–0.938]	5	0.713 [0.283–0.937]	7	**0.724** [0.559–0.890]	4
C2–C3	0.512 [0.181–0.934]	4	0.408 [0.025–0.661]	4	**0.500** [0.321–0.615]	6	0.340 [0.006–0.647]	4
C3–C4	0.511 [n/a]	1	0.412 [0.025–0.831]	5	0.382 [0.355–0.409]	2	**0.645** [n/a]	1
C4–C5	[n/a]	0	0.489 [0.464–0.514]	2	**0.578** [0.492–0.663]	2	0.281 [n/a]	1
C5–C6	[n/a]	0	0.605 [0.505–0.705]	2	0.535 [n/a]	1	**0.542** [0.314–0.772]	3
C6–C7	0.674 [n/a]	1	[n/a]	0	**0.770** [n/a]	1	0.685 [n/a]	1

ICC = Intraclass Correlation Coefficient. Per vertebra the highest mean score is indicated in red bold.

## Data Availability

The data presented in this study are available on request from the corresponding author due to privacy restrictions.

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
