# Peer review of "AI-Assisted Image Recognition of Cervical Spine Vertebrae in Dynamic X-Ray Recordings"

_bioengineering, 2025, doi:10.3390/bioengineering12070679_

Round 1
Reviewer 1 Report
Comments and Suggestions for Authors
The Artificial Intelligence Motion study (AIM): A feasibility study of AI-assisted image recognition of cervical spine vertebrae in dynamic X-ray recordings.
This paper explores the use of deep learning models, specifically a U-Net architecture, for segmenting cervical vertebrae in dynamic X-ray recordings. The study evaluates the feasibility of AI for analysing cervical spine motion by assessing the relative rotation of vertebrae during flexion and extension. Using a dataset from healthy individuals, the authors apply 2D and 2D+t models and report performance metrics such as DSC, IoU, and ICC. The results show the potential of AI in cervical spine motion analysis but highlight the need for further improvements in segmenting vertebral segments and optimizing the model for clinical applications.
Comments:
- The manuscript has multiple inconsistencies, such as figure references, and some spellings as well for example (in start of introduction section, the word “analyses”, can’t understand that if you wanted to write the analysis or analyzes). These issues disrupt the professionalism and readability of the paper. A thorough proofreading and formatting revision is necessary to address these issues (Throughout the paper).
- The methodology, specifically the use of a U-Net architecture for cervical spine vertebrae segmentation in dynamic X-ray recordings, is technically sound and well-justified. The authors have chosen an appropriate deep learning model (U-Net) for medical image segmentation. However, the paper could benefit from more clarity on the practical challenges faced during model training (e.g., data quality, choice of hyperparameters). A more in-depth explanation of the pre-processing steps would add scientific value (Section 2.3).
- The authors report reasonable values for Dice Similarity Coefficient (DSC) and Intersection over Union (IoU) but fail to sufficiently justify the choice of the thresholding technique for segmentation (e.g., IoU score calculation). Additionally, the performance of models A-D could be analyzed further by providing more detailed discussions of how these metrics correlate with clinical applicability. The presentation of ICC values for relative rotation is helpful, but deeper statistical analysis (e.g., confidence intervals) would help interpret these values in the context of clinical relevance (Section 3.4).
- The experimental design is generally solid, particularly the use of dynamic X-ray recordings to evaluate cervical spine motion. But the selection of specific image dimensions (640x640 vs. 832x576) could be more thoroughly discussed, especially considering the impact of these settings on model performance. It would be beneficial to include a comparison of the temporal dynamics between models (2D vs. 2D+t) with respect to actual clinical practice (Section 2.3, 3.1).
- While the study focuses on evaluating AI-assisted cervical vertebrae recognition, the biological relevance of motion patterns between vertebrae (e.g., segmental contributions) could be discussed more extensively. A better connection between the segmentation results and the clinical decision-making process is needed. For example, correlating the AI-segmented vertebrae with patient outcomes or surgical decisions would strengthen the implications of this study for real-world applications (Section 4).
- The authors briefly mention some limitations, such as the difficulty in segmenting lower vertebrae due to overlapping structures (e.g., shoulders), but a more structured discussion of limitations would be valuable. This section should include potential biases introduced by the model, challenges with segmenting certain vertebrae (C0, C7), and how these limitations affect model performance. More detailed suggestions for future model improvement (e.g., multi-modal data, incorporating axial rotation) would enhance the impact of the study (Section 4).
- The authors briefly touch upon future work, including improving model segmentation, but a more detailed roadmap for future research would be helpful. Suggestions for using multi-dimensional data, improving image quality through contrast enhancement, and extending the model to clinical datasets (patients with injuries or pathologies) would be valuable additions (Section 5).
- The conclusions are generally aligned with the findings, emphasizing the feasibility of AI-assisted cervical vertebrae analysis. The potential clinical applications, such as aiding decision-making in spinal surgery or injury diagnosis, should be highlighted more clearly. Future research could focus on expanding the dataset to include pathological conditions, which would help generalize the model for wider use (Section 5).
Reviewer 2 Report
Comments and Suggestions for Authors
This is a well-motivated and timely study addressing a relevant clinical problem — the automation of cervical spine motion analysis using deep learning, which could significantly improve diagnostic efficiency. The implementation of both 2D and 2D+t U-Net models, along with a clear evaluation strategy, strengthens the manuscript. However, several areas need further clarification and refinement in methodology, presentation, and interpretation of results.
- The title is informative and adequately reflects the content of the study. But the title should be shorter.
- The abstract mentions high sensitivity and specificity of motion analysis but does not quantify them or clearly link these terms to the deep learning models.
- The concluding sentence should include a concrete future direction or implication beyond general feasibility.
- The literature review could be better organized by separating traditional manual methods and recent AI-based approaches.
- Some references to earlier studies are vaguely introduced (e.g., references 6, 7, 12, 13).
- Details about the number of frames input into the 2D+t models and how temporal consistency was modeled are missing.
- The statistical basis for the dataset split and sample size sufficiency is not discussed.
- The explanation about the manual annotation is good, but one clarification is needed: how was the inter-rater agreement between the two annotators assessed?
- You mention "best-fit principle" — cite the specific algorithm or software used for this matching.
- All tables not in format, please rectify.
The English could be improved to more clearly express the research.
Round 2
Reviewer 1 Report
Comments and Suggestions for Authors
Most of my comments are addressed. I recommend the acceptance of this article.